METHODS

# Getting to GRIPS with MR-Egger: Modelling directional pleiotropy independently of allele coding

Frank Dudbridge[1]*, Bethany Voller[2], Ruby M. Woodward[1], Katie L. Saxby[3], Timothy M. Frayling[2,4], Luke C. Pilling[2], Jack Bowden[2,5]

1 Division of Public Health and Epidemiology, School of Medical Sciences, University of Leicester, Leicester, United Kingdom, 2 Department of Clinical and Biomedical Sciences, Faculty of Health and Life Sciences, University of Exeter, Exeter, United Kingdom, 3 Division of Cardiovascular Sciences, School of Medical Sciences, University of Leicester, Leicester, United Kingdom, 4 Department of Genetic Medicine and Development, Faculty of Medicine, CMU, University of Geneva, Geneva, Switzerland, 5 Novo Nordisk Research Centre Oxford, Oxford, United Kingdom

* frank.dudbridge@leicester.ac.uk

## Abstract

Mendelian Randomisation Egger regression (MR-Egger) is a popular method for causal inference using single-nucleotide polymorphisms (SNPs) as instrumental variables. It allows all SNPs to have direct pleiotropic effects on the outcome, provided that those effects are independent of the effects on the exposure, known as the InSIDE assumption. However, the results of MR-Egger, and the InSIDE assumption itself, are sensitive to which allele is coded as the effect allele for each SNP. A pragmatic convention is to code the alleles with positive effects on the exposure, which has some advantages in interpretation but some statistical limitations. Here we show that if the InSIDE assumption holds under all-positive coding of the exposure effects, it cannot hold under all-positive coding of the pleiotropic effects, and argue that this undermines the soundness of MR-Egger. We propose a modification that has the Genotype Recoding Invariance Property (GRIP), achieving the main aim of MR-Egger without the difficulties of allele coding. Our approach, MR-GRIP, is valid under a "Variance Independent of Covariance Explained" assumption (VICE), which amounts to an inverse relationship between exposure effects and pleiotropic effects. Examples and simulations suggest that MR-GRIP can reconcile differences between MR-Egger and alternative methods.

## Author summary

Mendelian Randomisation (MR) is a statistical method that can distinguish causal relationships from statistical correlations, under certain assumptions. The principle is to use genetic markers, such as single-nucleotide polymorphisms

**Data availability statement:** All relevant data are within the manuscript and its Supporting Information files.

**Funding:** This work was funded by the Medical Research Council (grant number MC/MR/WO14548/1 to TF). The funder had no role in study design, data collection and analysis, decision to publish, or preparation of the manuscript.

**Competing interests:** The authors have declared that no competing interests exist.

(SNPs), as proxies for the causal variable. One version of MR, called MR-Egger, is very popular but has a serious drawback in that its results depend on how the SNPs are numerically encoded. We propose a modification that has the Genotype Recoding Invariance Property (GRIP), which avoids this problem whilst achieving the main aim of MR-Egger. We illustrate our approach, called MR-GRIP, in simulations and in real data examples including the effect of serum urate on coronary heart disease (CHD), the effect of body mass index on coronary artery disease, and the joint effects of plasma lipids on CHD. In each case, MR-GRIP gives plausible results, and in some cases, it appears to reconcile differences between MR-Egger and alternative methods for MR.

## Introduction

In two-sample Mendelian randomisation (MR) using multiple single-nucleotide polymorphisms (SNPs), the standard estimator of the causal effect is the inverse-variance weighted (IVW) mean of ratio estimates [1–3]. This is usually supplemented with sensitivity analyses that relax the instrumental variable (IV) assumptions in various ways. Mendelian Randomisation Egger regression (MR-Egger) is often performed as a method that allows all SNPs to have direct pleiotropic effects on the outcome, not acting via the exposure [4]. The IVW estimate assumes "balanced pleiotropy", by which such pleiotropic effects have mean zero. MR-Egger allows a non-zero mean, called "directional pleiotropy". Both IVW and MR-Egger require the InSIDE assumption (Instrument Strength Independent of Direct Effect) under which the pleiotropic effects are independent of the SNP-exposure effects. This intuitively corresponds to independence of the corresponding biological pathways.

The InSIDE terminology, referring to direct effects of instruments, is somewhat careless as it is genotypes that have effects, not SNPs. Thus, in discussing directional pleiotropy, one must specify a numerical coding for each of the three genotypes of each SNP. An additive model is usually assumed, which only requires specifying which of the two alleles is the effect allele for each SNP. Depending on this *allele coding*, directional pleiotropy may or may not be present, and the InSIDE assumption may or may not hold [5]. This is rather discomfiting, as our assumptions and inferences ought to reflect some state of nature rather than the data coding. However, MR-Egger can be very sensitive to the allele coding, while IVW is invariant to it. The convention is to code the alleles with positive effects on the exposure, with the InSIDE assumption then expressed relative to that coding [6]. This may be problematic if, in truth, the InSIDE assumption only holds under some other coding. Lin et al [5] have extensively studied the properties of MR-Egger under an unknown oracle coding, demonstrating increased potential for biased inference when the coding is mis-specified, while being unable to identify a reliable strategy for inferring the oracle coding.

The concept of an allele coding itself needs refining since each SNP has different alleles. If the choice of effect allele is considered random, then there is little reason

not to assume balanced pleiotropy, as pleiotropic effects are as likely to be positive as negative [7]. For directional pleiotropy to have meaning, there must be a systematic way to code SNPs irrespective of their alleles. All-positive coding is one such scheme, although it was introduced only to standardise MR-Egger, not from biological considerations. Nevertheless, the existence of such a coding admits the possibility of directional pleiotropy, which could then be accommodated by MR-Egger.

In this paper we review the problem of defining the effect alleles, and how this can affect MR assumptions and results. We present a new argument to suggest that the all-positive coding of MR-Egger is logically problematic. We propose an alternative formulation of directional pleiotropy, MR-GRIP, which is invariant to allele coding. Using simulations and data examples, we show that this approach has promise as a sensitivity analysis for MR that achieves the main aim of MR-Egger while avoiding some of its difficulties.

## Description of the method

### Preliminaries

Assume the setting of a MR analysis, with the following data generating model for the continuous exposure $X$, outcome $Y$ and confounder $U$ for subject $i = 1, \cdots, n$, with SNP genotypes $G_{ij}$, $j = 1, \cdots, m$:

$$U_i | G_{i.} = \sum_{j=1}^{m} \phi_j G_{ij} + \in_i^U \tag{1}$$

$$X_i | U_i, G_{i.} = \sum_{j=1}^{m} \gamma_j G_{ij} + U_i + \in_i^X \tag{2}$$

$$Y_i | X_i, U_i, G_{i.} = \sum_{j=1}^{m} \alpha_j G_{ij} + \beta X_i + \delta U_i + \in_i^Y \tag{3}$$

Assume that the errors $\in_i^U$, $\in_i^X$ and $\in_i^Y$ are independent of all other variables in the above model, which is represented graphically in Fig 1.

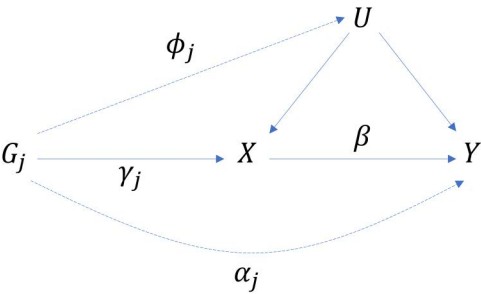

**Fig 1. Directed acyclic graph showing causal relations between gene $G_j$, exposure X, confounder U and outcome Y.** Lines are labelled with parameters from equations 1, 2 and 3. Dashed lines are absent under the standard instrumental variable assumptions.

This model is simplistic, and ignores many of the features that are present in real MR studies, such as binary outcomes, case/control sampling, non-additive effects and so on. Nevertheless, it serves as the basis for motivating many commonly used MR methods.

In MR the standard instrumental variable assumptions are usually stated as: $G$ associated with $X$ (IV1); $G$ independent of $U$ (IV2); $Y$ independent of $G$ given $X$ and $U$ (IV3). Assumptions IV1-IV3 are represented by the solid lines in Fig 1. The dashed lines represent examples of violations of these assumptions, as represented by the parameters $\phi_j$ and $\alpha_j$ in equations 1 and 3. As written, IV2 is violated for variant $j$ when $\phi_j \neq 0$ and IV3 is violated when $\alpha_j \neq 0$. In what follows we assume that $\phi_j = 0$ for all $j$, so that IV2 is always satisfied.

## Summary data MR

Often, only summary data estimates of genetic associations with the exposure and outcome are available for MR analysis. In this setting, a causal effect estimate may be obtained from each variant in turn via the ratio (or Wald) method, and the estimates may then be averaged by taking a (weighted) mean, median or mode. Marginalising over the confounder $U$ in equation 2 and over $U$ and $X$ in equation 3, equivalent models for $X|G_j$ and $Y|G_j$ (assuming all $\phi_j = 0$) are

$$X_i|G_{ij} = \gamma_j G_{ij} + \epsilon_i'^X \tag{4}$$

$$\begin{aligned} Y_i|G_{ij} &= (\alpha_j + \beta\gamma_j)G_{ij} + \epsilon_i'^Y \\ &= \Gamma_j G_{ij} + \epsilon_i'^Y \end{aligned} \tag{5}$$

Equations 4 and 5 are known as the "reduced form" equations in the econometrics literature. They furnish a ratio estimate for each variant $j$ as

$$\hat{\beta}_j = \frac{E(Y|G_j)}{E(X|G_j)} = \frac{\hat{\Gamma}_j}{\hat{\gamma}_j} = \frac{\widehat{(\alpha + \beta\gamma)}_j}{\hat{\gamma}_j} \tag{6}$$

Let $\sigma_{Xj}^2$, $\sigma_{Yj}^2$ denote the sampling variances of $\hat{\gamma}_j$, $\hat{\Gamma}_j$ respectively, assumed known. In line with other summary data methods [8] we assume that the SNPs are independent (*i.e.,* in linkage equilibrium) and have sufficiently small effects that, for $j \neq k$, the correlations between $\hat{\gamma}_j$ and $\hat{\gamma}_k$, and between $\hat{\Gamma}_j$ and $\hat{\Gamma}_k$, are negligible.

Summary data MR is most easily motivated by assuming that the standard error of the SNP-exposure association estimate is negligible ($\sigma_{Xj}^2 = 0$ for all $j$, so $\hat{\gamma}_j = \gamma_j$). This is tantamount to saying that each SNP is an infinitely strong instrument, and is called the NO Measurement Error (NOME) assumption because of the connection between the bias induced by weak instruments in MR and the general phenomenon of regression dilution bias. It also makes life easier if the SNP-exposure and SNP-outcome data are obtained from independent samples, so that $\epsilon_i'^X$ is independent of $\epsilon_i'^Y$ but the population is assumed to be homogeneous. This framework is called a two-sample MR analysis [9]. We begin by assuming the two-sample setting with NOME, relaxing these assumptions later.

## IVW and MR-Egger

The IVW approach to summary data MR obtains an overall estimate for the causal effect from the weighted mean of the individual ratio estimates (equation 6). Thus, it assumes that all SNPs are valid IVs, $\phi_j = \alpha_j = 0$ for all $j$. If so, the following simplified model holds:

$$\hat{\Gamma}_j = \beta\hat{\gamma}_j + \epsilon_{Yj}, \quad \epsilon_{Yj} \sim N(0, \sigma_{Yj}^2) \tag{7}$$

This is a linear regression model with the intercept constrained to be zero. With inverse variance weights $\sigma_{Yj}^{-2}$, the least squares estimate for the causal effect $\beta$ is

$$\hat{\beta}_{IVW} = \frac{\sum \sigma_{Yj}^{-2} \hat{\Gamma}_j \hat{\gamma}_j}{\sum \sigma_{Yj}^{-2} \hat{\gamma}_j^2} = \frac{\sum w_j \hat{\beta}_j}{\sum w_j} \tag{8}$$

where $w_j = \hat{\gamma}_j^2 \sigma_{Yj}^{-2}$ is the inverse variance of $\hat{\beta}_j$ under the NOME assumption. Under the IV and NOME assumptions, the IVW estimate is unbiased for $\beta$.

The IVW estimate is also unbiased for $\beta$ under pleiotropic effects $\alpha_j$ such that $E(\alpha_j|\gamma_j) = 0$, since equation (7) now becomes

$$\hat{\Gamma}_j = \beta \hat{\gamma}_j + \alpha_j + \in_{Yj} \tag{9}$$

and in fitting the regression the unobserved $\alpha_j$ are absorbed into the errors. The zero-mean assumption on $\alpha_j$ is known as balanced pleiotropy. The regression residuals must be independent of the exposure variable: that is, $\alpha_j + \in_{Yj}$ is independent of $\hat{\gamma}_j$, thus (in the two-sample design under NOME) we require independence of $\alpha_j$ and $\gamma_j$, known as the InSIDE assumption.

Under InSIDE we have $E(\alpha_j|\gamma_j) = E(\alpha_j)$. This expectation may be conceived as over the fixed effects of the specific SNPs in the analysis ("perfect" InSIDE [9]), or more usually, over a hypothetical sample space of random pleiotropic effects, whose variance contributes to the residual variance in the regression ("general" or "weak" InSIDE [9,10]). Balanced pleiotropy is more easily justified under the latter random effects conception, and we will adopt this view in the remainder of the article.

MR-Egger regression has been proposed as an extension to the IVW method without the constraint that the intercept equals zero [4]. It can, in theory, provide an unbiased estimate for the causal effect if, across variants, the pleiotropic effects $\alpha_j$ have non-zero expectation, "directional pleiotropy". The model in equation (9) is under-identified, as not all of its $(m + 1)$ parameters can be estimated simultaneously. MR-Egger regression resolves this by instead fitting the two-parameter model

$$\hat{\Gamma}_j = \alpha_0 + \beta \hat{\gamma}_j + \in_{Yj} \tag{10}$$

Similar to the IVW method, the $j$-th term of this regression is usually weighted by $\sigma_{Yj}^{-2}$ to improve efficiency, although this would only yield the correct standard error if all $\alpha_j = \alpha_0$. Two alternative approaches that impose a different identifying condition on equation (9) are the Weighted Median (WM) [11] and Mode-Based Estimator (MBE) [12]. They both assume that some of the instruments are not pleiotropic at all. In essence, this amounts to the condition that $median(\alpha_j) = 0$ in the case of the WM or that $mode(\alpha_j) = 0$ in the case of the MBE.

### Genotype Recoding Invariance Property (GRIP)

When conducting summary data MR, the SNPs are usually coded with an additive model to represent the number of exposure increasing alleles (0, 1 or 2). Therefore $\hat{\gamma}_j > 0$ for all $j$, which we call *all-positive coding*. However, the alternative allele could be coded instead, in which case $\hat{\gamma}_j$ and $\hat{\Gamma}_j$ both change sign. The coding is unimportant for the IVW estimate (equation 8), since the sign cancels in each ratio $\hat{\beta}_j$ (equation 6) and the weights $w_j$ are always positive. We say that the IVW estimate has the Genotype Recoding Invariance Property (GRIP); this property also applies to WM and MBE, and indeed to any method that works directly on the ratios $\hat{\beta}_j$.

A useful graphical interpretation of summary data MR is a scatter plot of the SNP-outcome association estimates $\hat{\Gamma}_j$ versus the SNP-exposure association estimates $\hat{\gamma}_j$. The ratio estimate $\hat{\beta}_j$ obtained from variant $j$ can then be interpreted as

the slope of the line linking the data point $\left(\hat{\Gamma}_j, \hat{\gamma}_j\right)$ to the origin. A line from the origin with slope $\hat{\beta}_{IVW}$ promotes its interpretation as a weighted average of the individual ratio estimates or slopes, obtained from the no-intercept regression model (equation 7).

To illustrate this, Fig 2a shows a hypothetical scatter plot of 14 SNP-exposure and SNP-outcome association estimates. The black dots represent the original associations of, say, the minor alleles. Five of the $\hat{\gamma}_j$ happen to be negative; the hollow dots show the transformation to all-positive coding, which is equivalent to a reflection of the negative values on the line $\hat{\Gamma} = -\hat{\gamma}$.

The IVW estimate, represented as the slope of the red line, is identical under either coding, since $\hat{\beta}_{IVW}$ has GRIP. However, the graphical interpretation of $\hat{\beta}_{IVW}$ as an average of individual slopes, and as a line of best fit through the data points, is only intuitive under all-positive coding. This is clarified in Fig 2b, which shows an extreme scenario of four SNP-exposure and SNP-outcome associations using the same conventions. Two of the variants yield negative ratio estimates and two yield positive ratio estimates that perfectly cancel, giving an IVW estimate of zero. The corresponding slope perfectly intersects the four data points under all-positive coding, but not under the original coding. Therefore, we may say that the IVW estimate has GRIP, but its scatter plot interpretation does not.

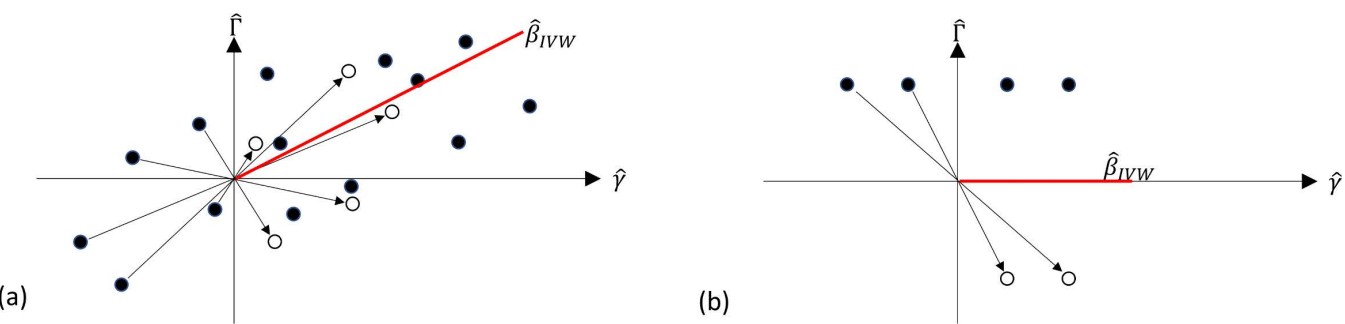

**Fig 2. Scatter plots of SNP-exposure association estimates $\hat{\gamma}$ and SNP-outcome association estimates $\hat{\Gamma}$ with original coding (solid dots) and all-positive coding (hollow dots).**

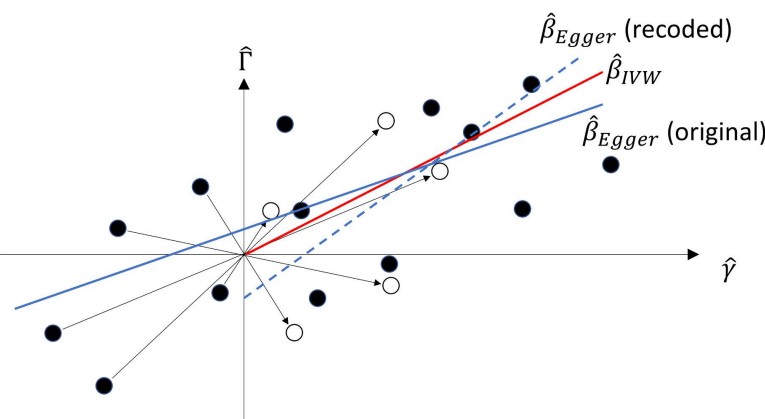

**Fig 3. Scatter plot of SNP-exposure association estimates $\hat{\gamma}$ and SNP-outcome association estimates $\hat{\Gamma}$ with original coding (solid dots) and all-positive coding (hollow dots).** MR-Egger slope for original data (solid blue line) and recoded data (dashed blue line). IVW slope for both (red line).

MR-Egger was originally proposed for all-positive coding, but the estimator does not have GRIP. In Fig 3, the solid red line is the IVW estimate, whereas the solid and dashed blue lines show the MR-Egger regression slope fitted to the original and all-positive coded data respectively. Inferences from the two MR-Egger estimates are markedly different: all-positive coding suggests negative directional pleiotropy and a causal effect greater than $\hat{\beta}_{IVW}$, whereas the original coding suggests positive directional pleiotropy and a causal effect less than $\hat{\beta}_{IVW}$. Indeed, each of the possible $2^{14}$ codings may lead to a different estimate of the causal effect.

This is surprising, because an equivalent statement of equation (9) is

$$-\hat{\Gamma}_j = \beta\left(-\hat{\gamma}_j\right) + \left(-\alpha_j\right) + \left(-\in_{Yj}\right) \tag{11}$$

so MR-Egger ought to estimate the same causal effect under all codings. The intercept, however, does depend on the coding, and thus any statement about directional or balanced pleiotropy is coding specific.

One reason for the lack of GRIP is the least-squares estimation of $\beta$. The weighted least-squares estimate in MR-Egger is

$$\frac{\sum \sigma_{Y_j}^{-2}\hat{\Gamma}_j\hat{\gamma}_j - \sum \sigma_{Y_j}^{-2}\hat{\Gamma}_j \sum \sigma_{Y_j}^{-2}\hat{\gamma}_j}{\sum \sigma_{Y_j}^{-2}\hat{\gamma}_j^2 - \left(\sum \sigma_{Y_j}^{-2}\hat{\gamma}_j\right)^2} \tag{12}$$

where the second terms in both numerator and denominator change under allele recoding. A more serious possibility, however, is that the InSIDE assumption itself does not have GRIP, and so MR-Egger is unbiased under some codings but not others. For example, Fig 4 shows a hypothetical scatter plot of 14 pleiotropic effects versus the SNP-exposure estimates. Under the original coding, $\alpha$ and $\hat{\gamma}$ are independent, satisfying InSIDE, whereas under all-positive coding a correlation is introduced and the assumption is violated.

It is common to test whether the intercept in MR-Egger is zero. Rejection of this hypothesis suggests that the IVW model may be inappropriate owing to directional pleiotropy. While the intercept generally depends on allele coding, the null hypothesis does have GRIP, because if $E(\alpha_j) = 0$ when $\gamma_i < 0$ then recoding also results in $E(-\alpha_j) = 0$. If there is a coding under which the null hypothesis $\alpha_0 = 0$ is true, it will remain true under any recoding (such as all-positive) that is independent of $\alpha$. Therefore, the intercept test is valid, in the sense of having the correct type-1 error, assuming InSIDE under balanced pleiotropy. Allele coding is only an issue if the null hypothesis is rejected, in which case an unbiased estimate of the causal effect requires the InSIDE assumption specific to all-positive coding.

An alternative representation of summary data MR is the Radial plot [13]. This plots the standardised estimates $\hat{\beta}_j/se(\hat{\beta}_j)$ against the inverse standard errors $1/se(\hat{\beta}_j)$, allowing easier identification of outliers: the slope of the corresponding

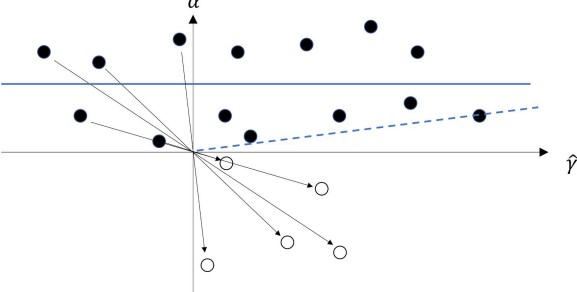

**Fig 4. Scatter plot of SNP-exposure association estimates $\hat{\gamma}$ and direct pleiotropic effects $\alpha$ with original coding (solid dots) and all-positive coding (hollow dots).** Regression lines for original data (solid blue line) and recoded data (dashed blue line).

regression (with zero intercept) is the IVW estimate and the distance from the slope to each SNP's data point exactly corresponds to its residual on a common scale. This may be extended to a Radial version of MR-Egger, via the regression model

$$\hat{\beta}_j \sqrt{w_j} = \alpha_0 + \beta \sqrt{w_j} + \in_{Yj} \tag{13}$$

where $\sqrt{w_j} = 1/se\left(\hat{\beta}_j\right)$. Because the standard error is naturally positive, and $\hat{\beta}_j$ is coding invariant, Radial MR-Egger ostensibly has GRIP. However, $se\left(\hat{\beta}_j\right) = \sqrt{var\left(\hat{\beta}_j\right)} = \sqrt{\hat{\gamma}_j^2/\sigma_{Y_j}^2} = |\hat{\gamma}_j| / |\sigma_{Y_j}|$, showing that the Radial model implicitly imposes all-positive coding. Extensive analyses have confirmed that results from all-positive MR-Egger and Radial MR-Egger are very similar [5].

## Pros and cons of all-positive coding

As a default for MR-Egger, all-positive coding has some good motivations. It allows an interpretation that all SNPs proxy the same intervention on the exposure. It reflects what might be done if a polygenic score were used as a single instrument, as it is natural to use all-positive coding in the score [3,14]. Furthermore, as noted above, the scatter plot interpretation makes most sense under all-positive coding.

One limitation is that the coding is based on sample estimates $\hat{\gamma}_j$ and if NOME is violated, some estimates may have different sign to the true $\gamma_j$. This should not be a problem when using SNPs strongly associated with the exposure, but it can lead to bias when many weak instruments are used [15]. To have any grounding in biology, the InSIDE assumption should apply to the true effects, but it may then be violated under all-positive coding of the sample estimates.

Another issue is that, by forcing all $\hat{\gamma}_j$ to be positive, their variance will be reduced compared to when some are negative. As a result, the estimator from all-positive coding will have greater standard error than those from other codings, since it is inversely proportional to the variability in $\hat{\gamma}_j$ [5,10]; thus, this default is, in a sense, the least efficient approach among those possible.

A more fundamental limitation of all-positive coding is that the InSIDE assumption may not be symmetric in $\alpha$ and $\gamma$. That is, if $\alpha_j$ and $\gamma_j$ are independent under the coding with $\gamma_j \geq 0$, they may not be independent under the coding with $\alpha_j \geq 0$. To see this, suppose that, under all-positive coding of $\gamma$, $\alpha < 0$ with probability $\pi$ and $\alpha \geq 0$ with probability $(1 - \pi)$. Define all-positive coding of $\alpha$ by $\alpha' = -\alpha$ and $\gamma' = -\gamma$ when $\alpha < 0$; $\alpha' = \alpha$ and $\gamma' = \gamma$ when $\alpha \geq 0$. Then (suppressing subscripts for clarity)

$$\begin{aligned}
cov\left(\alpha', \gamma'\right) &= E(\alpha'\gamma') - \left[\pi E\left(\alpha'\big|\alpha < 0\right) + (1-\pi)E\left(\alpha'\big|\alpha \geq 0\right)\right]\left[\pi E\left(\gamma'\big|\alpha < 0\right) + (1-\pi)E(\gamma'\big|\alpha \geq 0)\right] \\
&= E(\alpha\gamma) - \left[-\pi E\left(\alpha\big|\alpha < 0\right) + (1-\pi)E\left(\alpha\big|\alpha \geq 0\right)\right]\left[-\pi E\left(\gamma\big|\alpha < 0\right) + (1-\pi)E(\gamma\big|\alpha \geq 0)\right] \\
&= E(\alpha\gamma) - \left[E(\alpha) - 2\pi E(\alpha\big|\alpha < 0)\right]\left[E(\gamma) - 2\pi E(\gamma\big|\alpha < 0)\right] \\
&= cov(\alpha, \gamma) + 2\pi\left[E\left(\alpha\big|\alpha < 0\right)E(\gamma) + E(\alpha)E\left(\gamma\big|\alpha < 0\right)\right] - 4\pi^2 E(\alpha\big|\alpha < 0)E(\gamma\big|\alpha < 0)
\end{aligned} \tag{14}$$

Therefore, in general, $cov\left(\alpha', \gamma'\right) \neq cov(\alpha, \gamma)$ and they cannot both be zero. For example, in Fig 4, $\alpha$ and $\gamma$ are independent under all-positive coding of $\alpha$ but not under all-positive coding of $\gamma$. An exception to this asymmetry is balanced pleiotropy with $E(\alpha) = 0$ and $\pi = \frac{1}{2}$. Then the above reduces to

$$cov\left(\alpha', \gamma'\right) = cov(\alpha, \gamma) + E\left(\alpha\big|\alpha < 0\right)\left[E(\gamma) - E(\gamma\big|\alpha < 0)\right] \tag{15}$$

Under independence of $\alpha$ and $\gamma$, $E(\gamma) = E(\gamma|\alpha < 0)$ and $cov\left(\alpha', \gamma'\right) = cov(\alpha, \gamma) = 0$.

The asymmetry is disturbing because, in Fig 1, the only logical distinction between $\alpha$ and $\gamma$ is that $X$ has a causal effect on $Y$ but not *vice versa*. But the presence of this causal effect is just what we wish to infer. The all-positive InSIDE assumption bestows a status on $X$ that is not given to $Y$, but nature is presumably agnostic to what causal inferences we perform: there is no reason to expect that, if the all-positive InSIDE assumption does hold, it is in the same direction

as the causal inference of our interest. Similarly, a given set of SNPs can only be valid instruments for MR-Egger in one direction, despite satisfying assumptions IV1 and IV2 in both directions.

Notwithstanding the adage that all models are wrong, their assumptions should be justifiable by an argument from nature. For example, one could argue for balanced pleiotropy on the grounds that, if direct effects on $Y$ are indeed independent of those on $X$, they are equally likely to be positive or negative, and we have seen that InSIDE is symmetric in this case. Under directional pleiotropy, however, our view is that the all-positive InSIDE assumption is logically problematic owing to its structural asymmetry. Together with the statistical aspects noted above, this limitation severely undermines MR-Egger as the most natural generalisation of IVW.

## MR-GRIP and the VICE assumption

The issues related to all-positive coding are known, but they have been tolerated for want of a practical alternative. The prevailing view is that "Until a better solution appears, we should be cautious when applying MR-Egger" [5]. We now propose a simple modification that achieves the main aim of MR-Egger, estimating a mean of $\hat{\beta}_j$ with unbalanced pleiotropy, while retaining GRIP.

Recall equation (9)

$$\hat{\Gamma}_j = \beta\hat{\gamma}_j + \alpha_j + \in_{Yj} \tag{16}$$

Then trivially, multiplying through by $\hat{\gamma}_j$,

$$\hat{\Gamma}_j\hat{\gamma}_j = \beta\hat{\gamma}_j^2 + \alpha_j\hat{\gamma}_j + \in_{Yj}\hat{\gamma}_j \tag{17}$$

Our proposal, called MR-GRIP, is to estimate $\beta$ from the regression of $\hat{\Gamma}\hat{\gamma}$ on $\hat{\gamma}^2$. Clearly this regression has GRIP since recoding SNP $j$ changes the sign of both $\hat{\Gamma}_j$ and $\hat{\gamma}_j$ while $\hat{\gamma}_j^2$ is unchanged. Specifically, MR-GRIP fits the following mean model by linear regression:

$$E\left(\hat{\Gamma}_j\hat{\gamma}_j|\hat{\gamma}_j^2\right) = \alpha_0 + \beta\hat{\gamma}_j^2 \tag{18}$$

where the intercept is now $\alpha_0 = E(\alpha_j\hat{\gamma}_j)$ and may be zero ("balanced") or non-zero ("directional"). Still assuming NOME, the $\hat{\gamma}_j$ may be treated as fixed, in which case the variance of $\hat{\Gamma}_j\hat{\gamma}_j$ is $\hat{\gamma}_j^2\sigma_{Yj}^2$ and the regression of equation (18) may be fitted with inverse variance weighting as before. Explicit expressions for the weighted least-squares estimate and its standard error are given in S1 Text. In the special case that the intercept is fixed to zero, the estimate is

$$\frac{\sum \hat{\gamma}_j^{-2}\sigma_{Yj}^{-2}\left(\hat{\Gamma}_j\hat{\gamma}_j\right)\left(\hat{\gamma}_j^2\right)}{\sum \hat{\gamma}_j^{-2}\sigma_{Yj}^{-2}\left(\hat{\gamma}_j^2\right)^2} = \frac{\sum \sigma_{Yj}^{-2}\hat{\Gamma}_j\hat{\gamma}_j}{\sum \sigma_{Yj}^{-2}\hat{\gamma}_j^2} \tag{19}$$

identical to the standard IVW estimate. Thus, MR-GRIP can be seen as a generalisation of IVW that allows $E\left(\alpha_j\gamma_j|\gamma_j^2\right) \neq 0$.

Generally, from equation (17) the MR-GRIP estimate is

$$\frac{cov_w\left(\hat{\Gamma}_j\hat{\gamma}_j, \hat{\gamma}_j^2\right)}{var_w\left(\hat{\gamma}_j^2\right)} = \frac{cov_w\left(\beta\hat{\gamma}_j^2 + \alpha_j\hat{\gamma}_j + \in_{Yj}\hat{\gamma}_j, \hat{\gamma}_j^2\right)}{var_w\left(\hat{\gamma}_j^2\right)}$$

$$= \beta + \frac{cov_w\left(\alpha_j\hat{\gamma}_j, \hat{\gamma}_j^2\right)}{var_w\left(\hat{\gamma}_j^2\right)} + \frac{cov_w\left(\in_{Yj}\hat{\gamma}_j, \hat{\gamma}_j^2\right)}{var_w\left(\hat{\gamma}_j^2\right)} \tag{20}$$

where $cov_w$ and $var_w$ denote inverse-variance weighted covariance and variance respectively. The third term above is zero from the definition of $\in_{Yj}$. The MR-GRIP estimate is therefore unbiased if $cov_w\left(\alpha_j\hat{\gamma}_{lj}, \hat{\gamma}_j^2\right) = 0$, which is a counterpart to the InSIDE assumption [4]. It is sufficient that $\alpha_j\gamma_j$ and $\gamma_j^2$ are independent, and henceforth we will make that assumption. Similarly, to the InSIDE assumption, the independence may be assumed for the specific SNP effects in the analysis, or (more commonly) for a hypothetical space of random effects from which the SNP effects are drawn.

The converse assumption, independence of $\alpha_j\gamma_j$ and $\alpha_j^2$, is not guaranteed, but neither is it excluded by construction. This stands in contrast to MR-Egger, for which we showed above that in general, InSIDE cannot hold simultaneously for all-positive coding of $\gamma_j$ and for all-positive coding of $\alpha_j$. While the parameters $\alpha_j$ and $\gamma_j$ are natural in the linear model for a single SNP (equations 4 and 5), for inference across multiple SNPs we suggest that the canonical parameters are in fact the GRIP terms $\alpha_j\gamma_j$ and $\gamma_j^2$. These represent contributions to the direct genetic covariance of $X$ and $Y$ and the variance explained in $X$. We call the independence of $\alpha_j\gamma_j$ and $\gamma_j^2$ the VICE assumption (Variance Independent of Covariance Explained), which is sufficient for unbiased estimation by MR-GRIP.

Suppose that InSIDE holds under all-positive coding of $\gamma$, as assumed by MR-Egger. Then

$$
\begin{aligned}
cov\left(\alpha\gamma, \gamma^2\right) &= E\left(\alpha\gamma^3\right) - E(\alpha\gamma)E(\gamma^2) \\
&= E(\alpha)\left[E\left(\gamma^3\right) - E(\gamma)E\left(\gamma^2\right)\right] \\
&= E(\alpha)cov(\gamma, \gamma^2)
\end{aligned}
\tag{21}
$$

Since $\gamma > 0$, the VICE assumption can only hold if $E(\alpha) = 0$. Therefore, the assumptions of MR-Egger and MR-GRIP are only compatible under balanced pleiotropy, as assumed by IVW.

Similar to MR-Egger, the intercept test of the null hypothesis $\alpha_0 = 0$ may be used to infer directional pleiotropy, now defined as $E\left(\alpha_j\gamma_j\right) \neq 0$. We have shown above that the VICE assumption holds under the IVW assumptions of InSIDE under balanced pleiotropy; therefore, the MR-GRIP intercept test has the correct type-1 error under those conditions. Similar to MR-Egger, on rejecting balanced pleiotropy we require the VICE assumption for unbiased estimation of the causal effect.

If, however, $\alpha$ and $\gamma$ are not assumed to be independent, we can only allow the following relationship:

$$
\alpha = \frac{(\alpha\gamma)}{\sqrt{(\gamma^2)}} = \frac{\alpha_0}{\sqrt{(\gamma^2)}} + \frac{(\alpha\gamma - \alpha_0)}{\sqrt{(\gamma^2)}}
\tag{22}
$$

Under VICE, $\alpha\gamma$ and $\alpha_0$ are independent of $\gamma^2$. The first term therefore implies that $E(\alpha)$ is proportional to $\gamma^{-1}$ and the second that $\alpha$ varies stochastically with $\gamma^{-1}$. An example is shown in Fig 5d below. This relationship may be plausible if stronger effects on exposure $X$ are more specific to $X$, as might be the case for example if $X$ is a protein with strong instruments comprised of *cis* variants and weaker instruments from more dispersed *trans* signals [16]. Therefore, we suggest that MR-GRIP is compatible with a plausible biological model.

## Weak instruments

When NOME is violated, we have $\hat{\gamma}_j = \gamma_j + \in_{Xj}$ where we assume $\in_{Xj} \sim N(0, \sigma_{Xj}^2)$. In two-sample MR, large values of $\sigma_{Xj}^2$ create weak instrument bias towards the null. Similar to previous work for IVW models [15,17], we propose a bias adjusted estimator for $\beta$ by expressing the unobserved estimator under NOME in terms of the observed quantities $\hat{\gamma}_j$ and $\sigma_{Xj}^2$. For fixed weights $w_j$, assumed independent of $\gamma_j$,

$$
\hat{\beta}_{NOME} = \frac{\sum w_j \sum w_j \hat{\Gamma}_j \gamma_j^3 - \sum w_j \hat{\Gamma}_j \gamma_j \sum w_j \gamma_j^2}{\sum w_j \sum w_j \gamma_j^4 - \left(\sum w_j \gamma_j^2\right)^2}
\tag{23}
$$

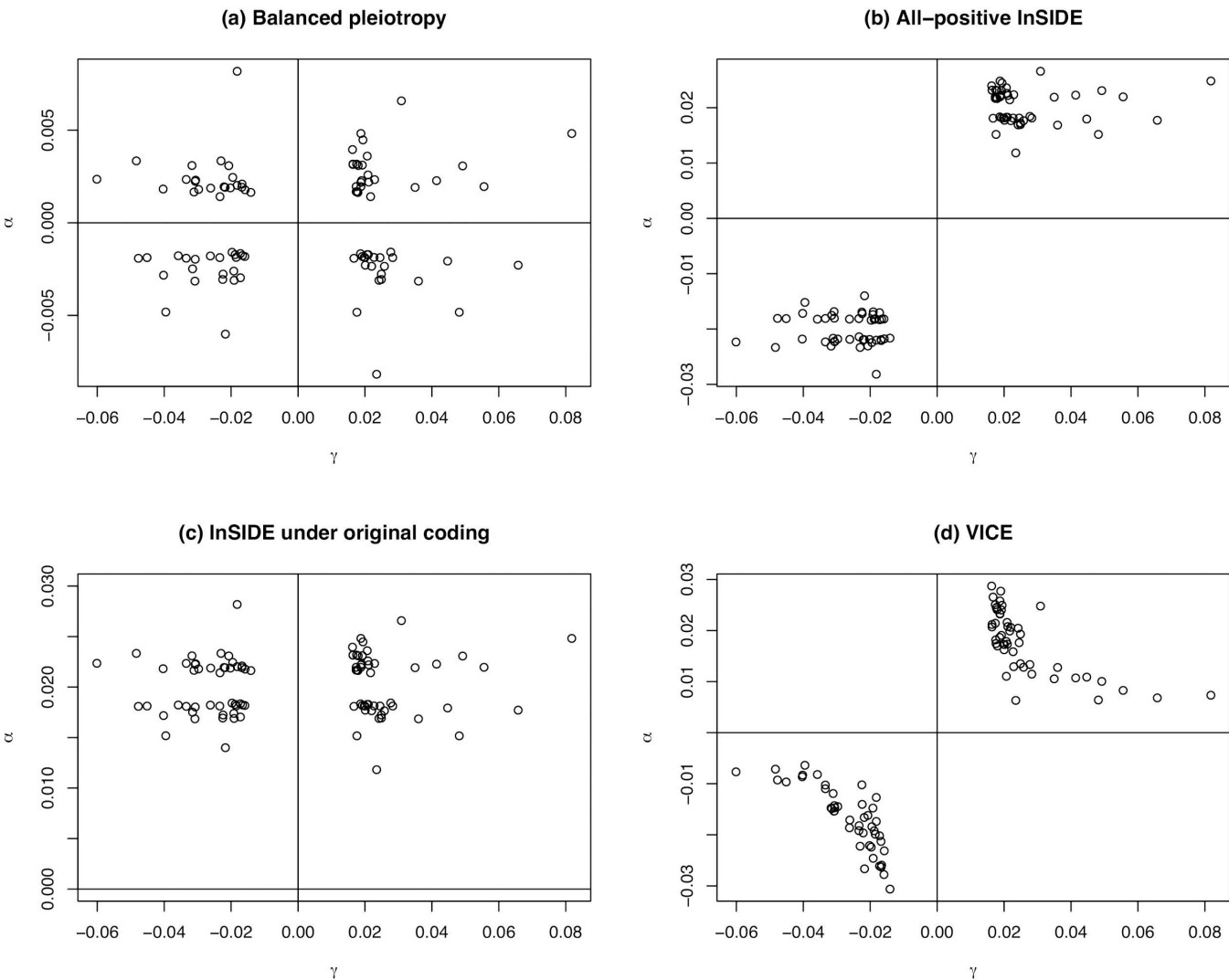

**Fig 5. Relationship between SNP effects on exposure $\gamma$ and direct pleiotropic effects $\alpha$ in one simulation based on the BMI example. a) Scenario 1; b) Scenario 2; c) Scenario 3; d) Scenario 4.**

where in the numerator

$$\sum w_j \hat{\Gamma}_j \gamma_j^3 \approx \sum w_j \hat{\Gamma}_j \hat{\gamma} \left( \hat{\gamma}_j^2 - 3\sigma_{Xj}^2 \right) \tag{24}$$

$$\sum w_j \hat{\Gamma}_j \gamma_j \approx \sum w_j \hat{\Gamma}_j \hat{\gamma}_j \tag{25}$$

$$\sum w_j \gamma_j^2 \approx \sum w_j \left( \hat{\gamma}_j^2 - \sigma_{Xj}^2 \right) \tag{26}$$

and in the denominator

$$\sum w_j \gamma_j^4 \approx \sum w_j \left( \hat{\gamma}_j^4 - 6\hat{\gamma}_j^2 \sigma_{Xj}^2 + 3\sigma_{Xj}^4 \right) \tag{27}$$

A derivation, including standard error, is provided in S1 Text. Note that the odd powers of $\gamma_j$ in $\hat{\beta}_{NOME}$ appear in products with $\hat{\Gamma}_j$ so that the estimator still has GRIP. However, the inverse variance weights $\hat{\gamma}_j^{-2} \sigma_{Yj}^{-2}$ are not independent of $\gamma_j$ and cannot be used with this approximation. Instead, we suggest taking $w_j = \sigma_{Yj}^{-2}$ as in standard IVW and MR-Egger, sacrificing some efficiency in estimating $\beta$.

## Variations

Here we indicate how the GRIP principle can be applied to some variants of MR-Egger. Firstly, recall the Radial model in which $\hat{\beta}_j \sqrt{w_j}$ is regressed on $\sqrt{w_j}$, where $\hat{\beta}_j = \hat{\Gamma}_j / \hat{\gamma}_j$ and $w_j = \hat{\gamma}_j^2 / \sigma_{Yj}^2$, imposing all-positive coding if positive square roots are taken. A GRIP version may be defined as the regression of $\hat{\beta}_j w_j$ on $w_j$, which could be extended to higher-order weights [13]. However, the graphical interpretation of the Radial plot, useful for identifying outlying $\hat{\beta}_j$, would not apply here and it is not clear what advantage a Radial MR-GRIP would otherwise have.

Secondly, consider multivariable MR for $k$ exposures $X_1, \cdots, X_k$ [18]. The summary statistic model for SNP $j$ is

$$\hat{\Gamma}_j = \beta_1 \hat{\gamma}_{1j} + \beta_2 \hat{\gamma}_{2j} + \cdots + \beta_k \hat{\gamma}_{kj} + \alpha_j + \in_{Yj} \tag{28}$$

Therefore

$$\hat{\Gamma}_j \hat{\gamma}_{1j} = \beta_1 \hat{\gamma}_{1j}^2 + \beta_2 \hat{\gamma}_{2j} \hat{\gamma}_{1j} + \cdots + \beta_k \hat{\gamma}_{kj} \hat{\gamma}_{1j} + \alpha_j \hat{\gamma}_{1j} + \in_{Yj} \hat{\gamma}_{1j} \tag{29}$$

and we may estimate the causal effects from the multiple regression of $\hat{\Gamma}_j \hat{\gamma}_{1j}$ on $\hat{\gamma}_{1j}^2$, $\hat{\gamma}_{2j} \hat{\gamma}_{1j}, \cdots, \hat{\gamma}_{kj} \hat{\gamma}_{1j}$. The VICE assumption will be that $\alpha_j \gamma_{1j}$ is independent of $\gamma_{1j}^2$, $\gamma_{1j} \gamma_{2j}, \cdots, \gamma_{1j} \gamma_{kj}$ for each $j$. Here we have arbitrarily multiplied through by $\hat{\gamma}_{1j}$, but any of the $\hat{\gamma}_{2j}, \cdots, \hat{\gamma}_{kj}$ could be used, with each yielding a different estimate with corresponding VICE assumption. These estimates could be combined, with further assumptions, or perhaps more usefully, inspected individually as an element of sensitivity analysis.

Next, the collider-correction framework allows two-sample MR methods to be applied to the one-sample design [19]. First regress $Y$ on both $X$ and $G_j$ giving coefficient estimates $\hat{\beta}^*$ and $\hat{\alpha}_j^*$ respectively. The estimated causal effect is then $\hat{\beta} = \hat{\beta}^* + \hat{\delta}$ where $\hat{\delta}$ is obtained from the linear regression of $\hat{\alpha}_j^*$ on $\hat{\gamma}_j$. A GRIP version of collider-correction obtains $\hat{\delta}$ from the regression of $\hat{\alpha}_j^* \hat{\gamma}_j$ on $\hat{\gamma}_j^2$.

Finally, MR methods have been applied to reduce selection bias in GWAS [15]. In the notation of Fig 1, the aim is to estimate $\alpha_j$ when the given data are the $\hat{\gamma}_j$ and the SNP associations with $Y$ conditional on $X$, denoted $\hat{\Gamma}_j^*$. Index effect regression gives the estimator $\hat{\alpha}_j = \hat{\Gamma}_j^* - \hat{b} \hat{\gamma}_j$ where $\hat{b}$ is obtained from the linear regression of $\hat{\Gamma}_j^*$ on $\hat{\gamma}_j$. A GRIP version of index effect regression instead obtains $\hat{b}$ from the regression of $\hat{\Gamma}_j^* \hat{\gamma}_j$ on $\hat{\gamma}_j^2$.

## Verification and comparison

We compared MR-GRIP, with and without weak instrument correction, to the IVW, MR-Egger, WM and MBE methods, using simulations based on data examples described below. The ADEMP framework for the simulations was as follows:

Aims: to compare the bias and precision of MR-GRIP to its nearest competitors under their respective assumptions.

Data-generation: to obtain realistic distributions of SNP-exposure effects, we used point estimates for the effect of serum urate on coronary heart disease (CHD) as described in the Applications section below. We then repeated the simulation using estimates for the effect of body mass index (BMI) on coronary artery disease (CAD) also described in

Applications. We simulated pleiotropic effects based on the same point estimates, under four scenarios described below, and used the standard errors from the two data examples to generate summary association statistics.

Estimands: the average causal effect of a unit shift intervention in $X$ on the outcome $Y$, and the standard error of its estimator. More precisely, the average causal effect is $\beta = E[Y(X = x)] - E[Y(X = x - 1)]$ where $Y(X = x)$ denotes the potential outcome $Y$ under the intervention that sets $X$ to $x$.

Methods: In fitting a mean model, IVW and MR-Egger have the same aim as MR-GRIP, and in that sense are the direct competitors. WM and MBE methods are often performed alongside MR-Egger as sensitivity analyses for IVW, and are complementary in taking the different kinds of average. Many other MR methods can be interpreted as a form of mean, median or mode. As more advanced mean models could in principle include our GRIP approach, our interest is in comparing only the most basic methods.

Performance measures: empirical bias and standard deviation of the estimated causal effect. We also assessed analytic standard errors by comparison to empirical standard deviations of the point estimates.

In each simulation, estimates $\hat{\gamma}_j$ were simulated from $N(\gamma_j, \sigma^2_{Xj})$ where $\gamma_j$ were point estimates in the data examples and $\sigma^2_{Xj}$ the corresponding squared standard errors. A proportion $p \in \{0.3, \ 1\}$ of the SNPs were invalid instruments. We set the causal effect to $\beta = 0.2$ but obtained qualitatively similar results, not shown, with other values of $\beta$ including the null.

*Scenario 1:* balanced pleiotropy, InSIDE satisfied. Invalid instruments have pleiotropic effects $\alpha$ sampled with replacement from $\{\pm\gamma_1, \cdots, \pm\gamma_m\}$ and scaled by 0.1. The marginal SNP effects on outcome $Y$ are $\Gamma_j = \beta\gamma_j + \alpha_j$. All methods are expected to be unbiased.

*Scenario 2:* directional pleiotropy, InSIDE satisfied under all-positive coding. As for Scenario 1 except that $\Gamma_j = \beta\gamma_j + \alpha_j + \alpha_0$ where $\alpha_0 = 0.02$. MR-Egger is expected to be unbiased, as are the weighted median and mode for $p = 0.3$.

*Scenario 3:* directional pleiotropy, InSIDE satisfied under other coding. As for Scenario 2 except that the $\hat{\gamma}_j$ are oriented in the (apparently random) direction in the original publications. All methods are expected to be biased, except for the weighted median and mode for $p = 0.3$.

*Scenario 4:* directional pleiotropy, VICE satisfied. For invalid instruments, products $(\alpha_j\gamma_j)$ are sampled with replacement from $\{\gamma_1^2, \ \gamma_1\gamma_2, \ \cdots, \ \gamma_1\gamma_m, \ \gamma_2\gamma_1, \ \gamma_2^2, \ \cdots, \gamma_2\gamma_m, \cdots, \ \gamma_m\gamma_1, \ \gamma_m\gamma_2, \ \cdots, \gamma_m^2\}$ and scaled by 0.1. The marginal SNP effects are $\Gamma_j = \beta\gamma_j + (\alpha_j\gamma_j + \alpha_0^2)/\gamma_j$ where $\alpha_0 = 0.02$ as in Scenario 2. MR-GRIP is expected to be unbiased, as are the weighted median and mode for $p = 0.3$.

Fig 5 shows the relationship between the effects on exposure $\gamma_j$ and the pleiotropic effects $\alpha_j$ in each scenario.

In all scenarios we simulated estimates $\hat{\Gamma}_j$ from $N(\Gamma_j, \sigma^2_{Yj})$ where $\sigma^2_{Yj}$ were the squared standard errors in the data examples. For both the urate and the BMI settings, we performed 10,000 simulations for $p \in \{0.3, \ 1\}$ in each scenario, and performed IVW, MR-Egger, WM and MBE calculations using default settings in the TwoSampleMR package [20].

Mean estimates of the causal effect for the urate-based simulation are shown in Table 1. Qualitatively similar results were observed for the BMI-based simulation; results are provided in S1 Text. As expected, all methods are unbiased under Scenario 1 (balanced pleiotropy). Under Scenario 2 (all-positive InSIDE), only MR-Egger is unbiased. MR-GRIP shows less bias than IVW, and similar bias to WM and MBE. Under Scenario 3 (InSIDE under original coding), IVW is the least biased, whereas MR-GRIP shows less bias than MR-Egger. Again, MR-GRIP has a similar bias to WM and MBE, which show some bias even with only 30% invalid instruments owing to sampling errors in the summary statistics. Under Scenario 4 (VICE) both IVW and MR-Egger are biased while MR-GRIP is unbiased as expected; WM and MBE appear empirically unbiased. Overall, in terms of bias, MR-GRIP performs similarly to WM and MBE, and of the mean-based estimators it is intermediate between IVW and MR-Egger in each scenario.

Mean analytic standard errors and empirical standard deviations are shown in Table 2. Under Scenario 1, all standard errors are accurately estimated. Under Scenario 2, standard errors appear over-estimated for IVW, and under Scenarios 3 and 4, all methods appear to over-estimate their standard errors. Throughout, and similarly to the bias, the standard error of MR-GRIP is intermediate between IVW and MR-Egger, and similar to WM and MBE.

**Table 1. Mean estimates of $\beta = 0.2$ with 31 SNPs as instruments. SNP-exposure effects as for the urate data example (Table A in S1 Data). Scenarios described in main text. Proportion invalid, proportion of SNPs with direct pleiotropic effects on outcome. MR-GRIP weak, MR-GRIP with adjustment for weak instruments.**

| Scenario | Proportion invalid | IVW | MR-Egger | MR-GRIP | MR-GRIP weak | Weighted median | Mode-based estimator |
|----------|--------------------|-----|----------|---------|--------------|-----------------|----------------------|
| 1 | 0.3 | 0.199 | 0.198 | 0.198 | 0.200 | 0.200 | 0.200 |
| | 1 | 0.199 | 0.198 | 0.198 | 0.200 | 0.200 | 0.200 |
| 2 | 0.3 | 0.250 | 0.198 | 0.221 | 0.215 | 0.218 | 0.217 |
| | 1 | 0.374 | 0.198 | 0.280 | 0.252 | 0.274 | 0.273 |
| 3 | 0.3 | 0.207 | 0.218 | 0.213 | 0.217 | 0.214 | 0.214 |
| | 1 | 0.227 | 0.271 | 0.252 | 0.258 | 0.253 | 0.252 |
| 4 | 0.3 | 0.216 | 0.188 | 0.199 | 0.200 | 0.202 | 0.201 |
| | 1 | 0.256 | 0.162 | 0.201 | 0.200 | 0.207 | 0.207 |

**Table 2. Mean analytic standard errors, and empirical standard deviations of point estimates in simulations of Table 1.**

| Scenario | Proportion invalid | Standard error | IVW | MR-Egger | MR-GRIP | MR-GRIP weak | Weighted median | Mode-based estimator |
|----------|--------------------|----------------|-----|----------|---------|--------------|-----------------|----------------------|
| 1 | 0.3 | Analytic | 0.0259 | 0.0370 | 0.0291 | 0.0338 | | |
| | | Empirical | 0.0257 | 0.0367 | 0.0297 | 0.0338 | 0.0324 | 0.0314 |
| | 1 | Analytic | 0.0299 | 0.0426 | 0.0343 | 0.0396 | | |
| | | Empirical | 0.0297 | 0.0424 | 0.0341 | 0.0390 | 0.0373 | 0.0360 |
| 2 | 0.3 | Analytic | 0.0336 | 0.0462 | 0.0373 | 0.0450 | | |
| | | Empirical | 0.0285 | 0.0454 | 0.0359 | 0.0418 | 0.040 | 0.038 |
| | 1 | Analytic | 0.0433 | 0.0427 | 0.0360 | 0.0632 | | |
| | | Empirical | 0.0298 | 0.0422 | 0.0343 | 0.0392 | 0.0371 | 0.0377 |
| 3 | 0.3 | Analytic | 0.0346 | 0.0493 | 0.0398 | 0.0457 | | |
| | | Empirical | 0.0320 | 0.0450 | 0.0364 | 0.0414 | 0.0396 | 0.0391 |
| | 1 | Analytic | 0.0534 | 0.0762 | 0.0615 | 0.0696 | | |
| | | Empirical | 0.0300 | 0.0426 | 0.0351 | 0.0391 | 0.0377 | 0.0377 |
| 4 | 0.3 | Analytic | 0.0292 | 0.0413 | 0.0327 | 0.0383 | | |
| | | Empirical | 0.0250 | 0.0351 | 0.0283 | 0.0305 | 0.0292 | 0.0282 |
| | 1 | Analytic | 0.0403 | 0.0527 | 0.0419 | 0.0541 | | |
| | | Empirical | 0.0295 | 0.0391 | 0.0313 | 0.0311 | 0.0298 | 0.0283 |

Tables 3 and E in S1 Text show that the intercept tests of MR-Egger and MR-GRIP performed similarly, with their *P*-values having correlations exceeding 0.8 and neither test dominating the other. Thus, the MR-GRIP intercept appears just as suitable as the MR-Egger intercept for detecting violations of the IVW assumptions.

We repeated the simulations, introducing weak instrument bias by multiplying the standard errors of the SNP-exposure effects by 5. This is equivalent to a 96% reduction in sample size and led to a mean *F*-statistic of 10 for the urate example and 2.27 for the BMI example. Mean point estimates are shown in Table 4; we only show scenario 1, where all methods would be unbiased with strong instruments. As expected, weak instruments bias the estimates towards the null, with MR-GRIP showing a similar level of bias to IVW. The weak instrument adjusted MR-GRIP showed much reduced bias. Weak instrument bias was greater in the BMI simulation, and the adjusted MR-GRIP showed numerical instability (Figs A and B in S1 Text) with some extreme outliers. The median of the adjusted MR-GRIP was 0.211, less biased than the mean, but the interquartile range (0.238) was large.

**Table 3. Power (at $P \leq 0.05$) of the intercept tests of MR-Egger and MR-GRIP in simulations of Table 1.** *P*-values obtained from the ratio of point estimate to analytic standard error, assuming a standard normal distribution. Correlation, Spearman correlation between *P*-values of the two tests.

| Scenario | Proportion invalid | MR-Egger | MR-GRIP | Correlation |
|----------|-------------------|----------|---------|-------------|
| 1 | 0.3 | 0.0497 | 0.0658 | 0.820 |
|   | 1 | 0.0612 | 0.0638 | 0.821 |
| 2 | 0.3 | 0.338 | 0.326 | 0.914 |
|   | 1 | 1.00 | 1.00 | 0.865 |
| 3 | 0.3 | 0.0612 | 0.0587 | 0.840 |
|   | 1 | 0.0287 | 0.0357 | 0.904 |
| 4 | 0.3 | 0.145 | 0.185 | 0.897 |
|   | 1 | 0.711 | 0.734 | 0.938 |

**Table 4. Mean estimates of $\beta = 0.2$ under balanced pleiotropy.** Weak instruments simulated by multiplying the standard errors $\sigma_{X_j}$ of SNP-exposure effects $\hat{\gamma}_j$ by 5.

| Simulation | Proportion invalid | IVW | MR-Egger | MR-GRIP | MR-GRIP weak | Weighted median | Mode-based estimator |
|------------|-------------------|-----|----------|---------|--------------|-----------------|---------------------|
| Urate | 1 | 0.178 | 0.180 | 0.177 | 0.204 | 0.190 | 0.179 |
| BMI | 1 | 0.138 | 0.108 | 0.137 | 0.263 | 0.107 | 0.0956 |

## Applications

We illustrate MR-GRIP on some examples that have previously been used to compare MR methods. Firstly, the effect of plasma urate on CHD was historically used to elucidate MR-Egger. Similarly to Burgess and Thompson [6], 31 SNPs were used as instruments with effects on urate taken from the UCLEB consortium [21] and on CHD from the CARDIoGRAM-plusC4D consortium [22] (Table A in S1 Data). Secondly, the effect of BMI on CAD has been used to compare different approaches to pleiotropy in MR [23]. Following those authors, 97 SNPs were chosen as instruments with effects on BMI taken from the GIANT consortium [24]. Effects on CAD were available from the CARDIoGRAMplusC4D consortium [22] for 96 of these SNPs (Table B in S1 Data).

In both examples we calculated point estimates, standard errors and *P*-values. MR-GRIP, with and without weak instrument correction, was compared to the IVW, MR-Egger with all-positive and original coding, WM and MBE approaches.

Finally, the effects of low-density lipoprotein (LDL), high-density lipoprotein (HDL) and triglycerides have been considered in multivariate MR with CHD as the outcome [18]. Following those authors, 185 SNPs were used as instruments with effects on the three exposures taken from the Global Lipids Genetics Consortium [25]. Effects on CHD were again taken from CARDIoGRAMplusC4D [22] with data available for 182 SNPs (Table C in S1 Data). We calculated point estimates, standard errors and *P*-values for each exposure, using multivariate IVW, multivariate MR-Egger and multivariate MR-GRIP. We performed multivariate MR-Egger with all-positive coding of each exposure in turn. We performed multivariate MR-GRIP multiplying equation (9) by SNP-exposure effects for each exposure in turn.

For the urate example, the mean SNP *F*-statistic was 250.6, therefore we expected little bias due to weak instruments. Cochran's *Q* for heterogeneity was 89.3 on 30 d.f., suggesting substantial pleiotropy [26]. Estimated causal effects are shown in Table 5. IVW gives a statistically significant estimate whereas MR-Egger gives a null estimate with a statistically significant intercept test ($P = 0.008$). With the original coding, MR-Egger is similar to IVW. The other methods, including MR-GRIP, give similar intermediate estimates that are not nominally significant. The intercept in MR-GRIP is statistically significant ($P = 0.0006$), so it seems to reconcile the IVW with the WM and MBE results. The standard error of MR-GRIP

PLOS Genetics

**Table 5. Estimated causal effects of plasma urate on coronary heart disease. exp($\hat{\beta}$), estimated odds ratio per 1-sd increase in urate. se($\hat{\beta}$), standard error of log odds ratio.**

|  | exp$\left(\hat{\beta}\right)$ (95%CI) | se$\left(\hat{\beta}\right)$ | P |
|---|---|---|---|
| IVW | 1.11 (1.03 − 1.20) | 0.0401 | 0.00962 |
| MR-Egger | 1.00 (0.90 − 1.10) | 0.0515 | 0.980 |
| MR-Egger original | 1.12 (1.03 − 1.21) | 0.0395 | 0.00901 |
| MR-GRIP | 1.04 (0.96 − 1.13) | 0.0417 | 0.345 |
| MR-GRIP weak | 1.03 (0.95 − 1.12) | 0.0421 | 0.444 |
| Weighted median | 1.05 (0.99 − 1.11) | 0.0294 | 0.108 |
| Mode based estimator | 1.05 (1.00 − 1.11) | 0.0284 | 0.0776 |

is greater than IVW, WM and MBE, but lower than MR-Egger with all-positive coding. Overall, these results do not provide robust evidence for a causal effect of urate on CHD.

Fig 6a shows a scatter plot of the urate data, with all-positive coding of $\hat{\gamma}$ and the fitted models for IVW, MR-Egger and MR-GRIP. While the MR-GRIP model is linear in $\hat{\gamma}^2$, it is non-linear in $\hat{\gamma}$ since equation (18) implies

$$E\left(\hat{\Gamma}_{j}\Big|\hat{\gamma}_j\right) = \alpha_0\hat{\gamma}_j^{-1} + \beta\hat{\gamma}_j \tag{30}$$

While, by definition, this model is a better fit than IVW with $\alpha_0 = 0$, the graphical interpretation is less clear. For IVW and MR-Egger, the causal effect is the slope of the fitted line, whereas for MR-GRIP the causal effect is not apparent from the fitted curve. If the slope is interpreted as the causal effect, then it ostensibly varies with $\hat{\gamma}$. But in fact, the non-linear shape comes from the model for pleiotropic effects, which is added to the linear causal effect.

To allow a similar graphical interpretation to other MR methods, we suggest plotting a straight line on the scatter plot with slope set to the causal estimate from MR-GRIP, and intercept then estimated by weighted least squares. This gives a line that may be compared to the other methods, while appearing to fit the data. Similar to the MR-Egger line, the slope is the estimated causal effect, but it is not an average of slopes for each point. Fig 6b shows our proposed scatter plot for the urate data.

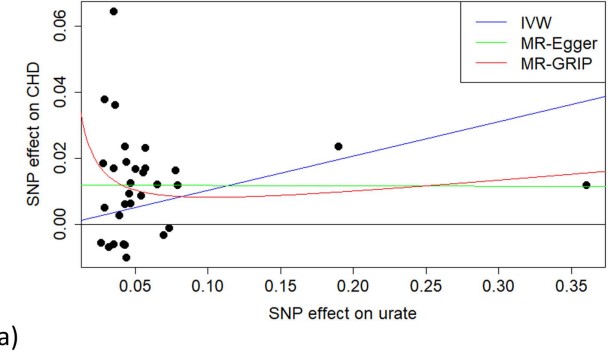

(a)

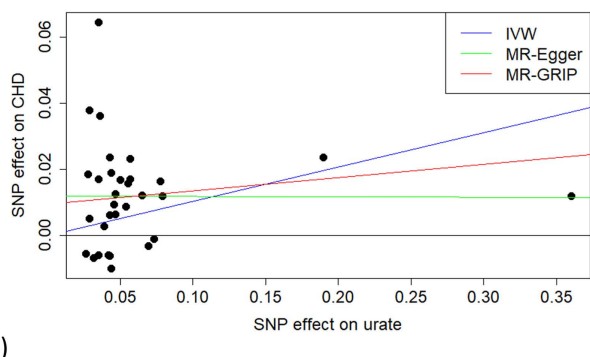

(b)

**Fig 6. Scatter plots of the urate data with fitted models.** a) non-linear MR-GRIP model. b) MR-GRIP represented by a straight line with slope equal to its causal estimate.

For the BMI example, the mean $F$-statistic was 56.7, therefore we again expected little bias due to weak instruments. Cochran's $Q$ for heterogeneity was 238.4 on 95 d.f., again suggesting substantial pleiotropy. Estimated causal effects are shown in Table 6. Here, MR-Egger gives a higher point estimate than IVW, but with reduced significance owing to its greater standard error. The intercept test gives weak evidence for directional pleiotropy ($P = 0.053$). Again, with the original coding MR-Egger is similar to IVW. MR-GRIP gives an intermediate estimate, consistent with the other methods, but there is no evidence for directional pleiotropy (intercept $P = 0.4$). MR-GRIP again has greater standard error than IVW, WM and MBE, but lower than MR-Egger. Weak instrument adjustment slightly increases the MR-GRIP estimate, but with increased standard error such that the evidence for causality is substantially weakened. As weak instrument bias is not expected, the adjustment seems unnecessarily imprecise. Overall the results suggest consistent evidence for a causal effect of BMI on CAD.

For the multivariate MR example, we give results for triglycerides in Table 7, as these include significant intercept tests for MR-Egger. Results for LDL and HDL are provided in S1 Text.

In univariate analysis, MR-GRIP agrees closely with IVW, with no evidence for directional pleiotropy. MR-Egger gives a reduced estimate with a significant intercept test. In multivariate analysis, the IVW estimate is attenuated, with MR-GRIP again in close agreement in all three versions. The results for MR-Egger are more problematic, with variation both in point estimates and in intercept tests depending on the trait coded all-positive. The standard guidance is to apply all-positive coding to the trait of primary interest [18], but here we may be led to different conclusions about triglycerides depending on whether it is a primary or secondary analysis. Furthermore, the InSIDE assumption cannot hold under all three codings; thus, we must make the dubious assumption that in nature, the InSIDE assumption holds under all-positive coding of the trait of our (but perhaps not another person's) primary interest.

These data examples suggest that MR-GRIP can give results that are compatible with those of other MR methods, and can help to resolve discrepancies between them.

## Discussion

MR-Egger was introduced to relax the balanced pleiotropy assumption of IVW. Since its inception it has been known that its results are sensitive to allele coding, complicating the interpretation of directional pleiotropy. More recently it been realised that the InSIDE assumption itself depends on allele coding. All-positive coding was introduced to standardise MR-Egger, with the InSIDE assumption applying specifically to that coding. If in fact the assumption holds under some other coding, MR-Egger may be severely biased, but it has proved difficult to identify an oracle coding from data [5].

While InSIDE under all-positive coding could exist in nature, it is more nuanced than the intuitive concept of independent biological pathways from SNP to exposure and to outcome. We have noted some limitations of all-positive coding, including inferring effect alleles from estimated effects, and performing multivariable MR. In addition, we have shown that

**Table 6. Estimated causal effects of BMI on coronary artery disease. $\exp(\hat{\beta})$, estimated odds ratio per 1-unit increase in BMI. $se(\hat{\beta})$, standard error of log odds ratio.**

| | $\exp\left(\hat{\beta}\right)$ (95% CI) | $se\left(\hat{\beta}\right)$ | $P$ |
|---|---|---|---|
| IVW | 1.44 (1.26 − 1.63) | 0.0648 | 2.5e-8 |
| MR-Egger | 1.61 (1.18 − 2.20) | 0.158 | 0.00323 |
| MR-Egger original | 1.42 (1.26 − 1.61) | 0.0640 | 3.1e-7 |
| MR-GRIP | 1.54 (1.25 − 1.89) | 0.105 | 8.5e-5 |
| MR-GRIP weak | 1.56 (1.07 − 2.26) | 0.190 | 0.0217 |
| Weighted median | 1.44 (1.25 − 1.66) | 0.0741 | 8.7e-7 |
| Mode- based estimator | 1.45 (1.20 − 1.75) | 0.0906 | 8.3e-5 |

**Table 7. Estimated causal effects of triglycerides on coronary heart disease.** $\exp(\hat{\beta})$, estimated odds ratio per 1-sd increase in inverse-normal transformed triglycerides. $se(\hat{\beta})$, standard error of log odds ratio. Coding, for MR-Egger, all-positive coding with respect to listed trait; for MR-GRIP, equation (9) multiplied by SNP-exposure effects for listed trait.

| | Coding | $\exp\left(\hat{\beta}\right)$ (95% CI)_ | $se(\hat{\beta})$ | P | Intercept P |
|---|---|---|---|---|---|
| **Univariate** | | | | | |
| IVW | | 1.36 (1.21 − 1.53) | 0.0600 | 2.3e-7 | |
| MR-Egger | | 1.14 (0.98 − 1.32) | 0.0743 | 0.0837 | 1.5e-4 |
| MR-GRIP | | 1.36 (1.21 − 1.53) | 0.0601 | 7.2e-7 | 0.473 |
| **Multivariate** | | | | | |
| IVW | | 1.17 (1.05 − 1.31) | 0.0549 | 0.00381 | |
| MR-Egger | LDL | 1.20 (1.07 − 1.34) | 0.0565 | 0.00137 | 0.111 |
| | HDL | 1.13 (1.01 − 1.26) | 0.0556 | 0.0325 | 0.00342 |
| | Triglycerides | 1.09 (0.96 − 1.23) | 0.0640 | 0.203 | 0.0205 |
| MR-GRIP | LDL | 1.18 (1.06 − 1.31) | 0.0548 | 0.00318 | 0.170 |
| | HDL | 1.17 (1.05 − 1.31) | 0.0552 | 0.00398 | 0.970 |
| | Triglycerides | 1.17 (1.05 − 1.31) | 0.0549 | 0.00382 | 0.298 |

under directional pleiotropy, InSIDE cannot hold simultaneously for all-positive coding of $\gamma$ and for all-positive coding of $\alpha$. Since, without knowledge of the causal direction, there is no logical distinction between these effects, we argue that this severely undermines the soundness of MR-Egger.

It could be held that for SNPs specifically chosen for an MR analysis, there is indeed a logical distinction between $\gamma$ and $\alpha$, such that $\gamma$ is expected to be larger in magnitude than $\Gamma = \beta\gamma + \alpha$ [27]. Then InSIDE under all-positive coding could hold for those SNPs, with no requirement for a symmetric condition, whereas for the reverse MR it could hold for a different set of SNPs. Whether such a scenario is plausible in nature is debatable, but this stance does allow MR-Egger to retain a place among MR methods. However, if faced with a discrepancy between MR-Egger and other methods, it would be bold to claim that the MR-Egger model is the more plausible. Thus, the practical usefulness of MR-Egger seems limited.

We propose MR-GRIP as an alternative model of directional pleiotropy, which does not depend on allele coding. With the intercept fixed to zero it is identical to IVW, so it achieves the same basic goal of MR-Egger – relaxing the balanced pleiotropy assumption in a mean model – without the difficulties raised by allele coding. The corresponding VICE assumption is compatible with InSIDE under balanced pleiotropy, so that the intercept may be tested similarly to MR-Egger. Under VICE, directional pleiotropy implies that $\alpha$ varies stochastically with $\gamma^{-1}$, which is plausible if we expect SNPS with strong effects on exposure to be more specific to that exposure, with reduced pleiotropic effects.

In simulations, MR-GRIP had bias and standard error that was intermediate between IVW and MR-Egger, and performed similarly to WM and MBE, especially the former. While we cannot offer theoretical explanations for these properties, these empirical findings are encouraging for the use of MR-GRIP in practice. Our simulations were necessarily limited as we were specifically interested in performance under the assumptions of each method. We did not consider violations such as outlier ratio estimates or correlated pleiotropy, as our aim was not to identify a preferred method over all possible scenarios, but to compare MR-GRIP to IVW and MR-Egger in the situations for which they are designed.

In two data examples, MR-GRIP gave estimates that were intermediate between IVW and MR-Egger. In the urate example, the intercept test was statistically significant, and the estimate was in line with WM and MBE. In the BMI example, despite not rejecting a zero intercept, the MR-GRIP estimate differed from IVW, WM and MBE, although confidence

intervals overlapped. We have observed similar results in other analyses to be reported elsewhere, although we are still early in our experiences.

In a multivariate MR example, MR-Egger gave estimates and intercept tests that varied according to which trait was coded all-positive. While MR-GRIP can also be implemented with different reference traits, in this example the results were consistent and in agreement with multivariate IVW.

Weak instruments are a potential problem for MR-GRIP. The degree of weak instrument bias appears comparable to IVW, although we have been unable to derive the exact magnitude of bias. We may apparently retain the rule of thumb that a mean $F$-statistic of at least 10 ensures little bias, but a strong bias may be more difficult to correct. We have proposed a formula to adjust for weak instrument bias, which generally performed well in our simulations. However, we have not proved that this estimator is unbiased, and when weak instrument bias is greater it appears more prone to numerical instability. An improved adjustment is an important direction for further work. However, approaches based on likelihood [8] would be challenging as the random variables in equation (11) have product distributions. Another useful area of future work would be the extension of MR-GRIP to correlated SNPs.

Our approach of multiplying equation (9) through by $\hat{\gamma}_j$ is not the only possibility. One could, for example, divide through instead, giving $\beta$ as the intercept in the regression of $\hat{\beta}_j$ on $\hat{\gamma}_j^{-1}$. In preliminary studies, we found this approach to be much less precise than MR-GRIP. Alternatively, one could multiply through by any odd power of $\hat{\gamma}_j$. We have not explored this, other than to note that it would require less intuitive, and perhaps less plausible, counterparts to the VICE assumption.

In summary, MR-GRIP provides a generalisation of IVW that avoids difficult arguments about the InSIDE assumption under all-positive coding. The VICE assumption is compatible with the IVW assumptions and also with an inverse relationship between pleiotropic and exposure effects. MR-GRIP appears to give results that are compatible with other MR methods, and can resolve discrepancies between IVW and MR-Egger. It is easily implemented, and has been added to the TwoSampleMR package [20]. We suggest that it can be easily included in the sensitivity analyses that are routinely performed in MR investigations.

## Supporting information

**S1 Data. Table A.** SNP effects (beta) and standard errors (se) on plasma urate (exposure) and on coronary heart disease (outcome) with the allele coding as listed in the original publication. **Table B.** SNP effects (beta) and standard errors (se) on BMI (exposure) and on coronary artery disease (outcome) with the allele coding as per the consortium download associated with the original publication. **Table C.** SNP effects (beta), P-values (p) and standard errors (se) on LDL, HDL, Triglycerides and on coronary artery disease (outcome) with the allele coding and lipid effects from the Global Lipids Genetics Consortium and CAD effects from the CARDIoGRAMplusC4D Consortium.
(XLSX)

**S1 Text. Table A.** Estimated causal effects of LDL cholesterol on coronary heart disease. $\exp\left(\hat{\beta}\right)$, estimated odds ratio per 1-sd increase in inverse-normal transformed LDL. $se\left(\hat{\beta}\right)$, standard error of log odds ratio. Coding, for MR-Egger, all-positive coding with respect to listed trait; for MR-GRIP, equation (9) multiplied by SNP-exposure effects for listed trait. **Table B.** Estimated causal effects of HDL cholesterol on coronary heart disease. $\exp\left(\hat{\beta}\right)$, estimated odds ratio per 1-sd increase in inverse-normal transformed HDL. $se\left(\hat{\beta}\right)$, standard error of log odds ratio. Coding, for MR-Egger, all-positive coding with respect to listed trait; for MR-GRIP, equation (9) multiplied by SNP-exposure effects for listed trait. **Table C.** Mean estimates of $\beta = 0.2$ with 96 SNPs as instruments. SNP-exposure effects as for the BMI data example (Table B in S1 Data). Scenarios described in main text. Proportion invalid, proportion of SNPs with direct pleiotropic effects on outcome. MR-GRIP weak, MR-GRIP with adjustment for weak instruments. **Table D.** Mean analytic standard errors, and empirical standard deviations of point estimates in simulations of Table C in S1 Text. **Table E.** Power (at $P \leq 0.05$) of the intercept tests of MR-Egger and MR-GRIP in simulations of Table C in S1 Text. $P$-values obtained from the ratio of point

estimate to analytic standard error, assuming a standard normal distribution. Correlation, Spearman correlation between $P$-values of the two tests. **Fig A.** Boxplots of causal effect estimates (left) and standard errors (right) for the weak instrument adjusted MR-GRIP in the urate simulation. **Fig B.** Boxplots of causal effect estimates (left) and standard errors (right) for the weak instrument adjusted MR-GRIP in the BMI simulation.
(DOCX)

## Acknowledgments

This is a summary of independent research carried out at the NIHR Leicester Biomedical Research Centre (BRC) and the NIHR Exeter BRC. The views expressed are those of the authors and not necessarily those of the MRC, ESRC, EPSRC, NIHR or the Department of Health and Social Care. We thank Tom Palmer and Gib Hemani for implementing MR-GRIP in the TwoSampleMR package.

## Author contributions

**Conceptualization:** Frank Dudbridge, Jack Bowden.

**Funding acquisition:** Timothy M. Frayling.

**Investigation:** Frank Dudbridge, Bethany Voller, Ruby M. Woodward, Katie L. Saxby, Timothy M. Frayling, Luke C. Pilling, Jack Bowden.

**Methodology:** Frank Dudbridge, Jack Bowden.

**Software:** Frank Dudbridge.

**Writing – original draft:** Frank Dudbridge, Jack Bowden.

**Writing – review & editing:** Frank Dudbridge, Bethany Voller, Ruby M. Woodward, Katie L. Saxby, Timothy M. Frayling, Luke C. Pilling, Jack Bowden.

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
