## [Decision Letter · Decision Letter 0]

22 Jul 2025

PGENETICS-D-25-00735

Getting to GRIPS with MR-Egger: modelling directional pleiotropy independently of allele coding

PLOS Genetics

Dear Dr. Dudbridge,

Thank you for submitting your manuscript to PLOS Genetics. After careful consideration, we feel that it has merit but needs some clarifications. Therefore, we invite you to submit a revised version of the manuscript that addresses the points raised during the review process.

Please submit your revised manuscript within 60 days Sep 20 2025 11:59PM. If you will need more time than this to complete your revisions, please reply to this message or contact the journal office at plosgenetics@plos.org. Please include the following items when submitting your revised manuscript:

We look forward to receiving your revised manuscript.

Kind regards,

Wei Pan

Guest Editor

PLOS Genetics

Xiaofeng Zhu

Section Editor

PLOS Genetics

Aimée Dudley

Editor-in-Chief

PLOS Genetics

Anne Goriely

Editor-in-Chief

PLOS Genetics

**Additional Editor Comments:**

I appreciate the discussion on this important problem and the proposed method that is both simple and creative! I'd encourage the authors to address the issues raised by the reviewers, most of which can be easily clarified.

**Journal Requirements:**

1) Please provide an Author Summary. This should appear in your manuscript between the Abstract (if applicable) and the Introduction, and should be 150-200 words long. The aim should be to make your findings accessible to a wide audience that includes both scientists and non-scientists. Sample summaries can be found on our website under Submission Guidelines:

https://journals.plos.org/plosgenetics/s/submission-guidelines#loc-parts-of-a-submission

2) Your manuscript is missing the following sections: Author Summary, Verification and Comparison, Applications, and Supplementary Information. Please ensure that your article adheres to the standard Methods article layout and order of Abstract, Author Summary, Introduction, Description of the Method, Verification and Comparison, Applications, Discussion, Acknowledgements, References, and Supplementary Information. For details on what each section should contain, see our Methods article guidelines:

https://journals.plos.org/plosgenetics/s/submission-guidelines#loc-manuscript-organization.

5) Please amend your detailed Financial Disclosure statement. This is published with the article. It must therefore be completed in full sentences and contain the exact wording you wish to be published. Please ensure that the funders and grant numbers match between the Financial Disclosure field and the Funding Information tab in your submission form. Note that the funders must be provided in the same order in both places as well.

**Reviewers' comments:**

Reviewer's Responses to Questions

**Comments to the Authors:**

Reviewer #1: see the attachment

Reviewer #2: This paper addresses a noteworthy issue in MR: the sensitivity of MR-Egger regression to allele coding. To address this, the authors propose a novel method, MR-GRIP, which is designed to possess the Gene Recoding Invariance Property (GRIP). The problem is interesting and relevant to the field. However, several aspects of the manuscript require clarification and significant improvement before it can be considered for publication.

1. The issue of MR-Egger’s sensitivity to allele coding, along with related concerns, has been previously and extensively discussed in Lin et al. (2022) [1]. It would greatly strengthen the manuscript if the authors could discuss this prior work and clearly articulate how their method builds upon, differs from, or addresses the limitations highlighted in that paper.

2.The statement that “the scatter plot and regression interpretations only make sense under all-positive coding” is confusing. Is Figure 2(b) just an extreme scenario?

3. The authors may wish to clarify that although the point estimate from IVW regression is invariant to gene recoding (i.e., satisfies GRIP), its assumptions—notably balanced pleiotropy and the InSIDE assumption—are not. This contrasts with WM and MBE, for which both the estimators and their assumptions are recoding-invariant. This distinction should be made more explicit in the text.

4. The statement that “the intercept test is valid in the sense of having the correct type-1 error, but if the null hypothesis is rejected then the causal effect estimate requires the InSIDE assumption specifically under all-positive coding” is unclear. Is the intercept test valid even when InSIDE does not hold? The authors should clarify under what conditions the test remains valid and how the coding impacts this validity.

5. The authors mention that “the InSIDE assumption may not be symmetric in \alpha and \gamma,” but this point is insufficiently explained. In particular, in "An important exception is balanced pleiotropy with E(\alpha) = 0 and \pi = 1/2", does the balanced pleiotropy refer to all-positive coding of \gamma? And why is E(\gamma)=E(\gamma|\alpha<0)?

6. The authors should write down the MR-GRIP model more explicitly, along with its assumptions. Based on my understanding, MR-GRIP is doing something like:

\hat\Gamma_j\hat\gamma_j = \beta\hat{\gamma_j}^2 + \alpha_0 + e_j. However, according to Eq (11), the error term is correlated with the predictor \hat\gamma_j^2, violating a core assumption of standard regression models. How does MR-GRIP address this issue? It is not clear that the VICE assumption fully accounts for this correlation. Also, how to interpret the intercept in MR-GRIP?

7. Could the authors clarify this statement "excepting balanced pleiotropy, the assumptions of MR-Egger and MR-GRIP are incompatible.". Is “balanced pleiotropy” defined with respect to an unknown or latent coding under which InSIDE holds?

8. The statement that “Because VICE holds under the IVW assumptions, the intercept test in MR-GRIP is a valid test of the IVW model” is ambiguous. What exactly is meant by “a valid test of the IVW model”? Does rejection imply model misspecification, or simply a violation of balanced pleiotropy? Also, the phrase “with rejection suggesting that MR-GRIP is a better fit to the data than the IVW model” should be clarified—how is “better fit” being defined in this context?

9. The part about "then under VICE we can only allow the following relationship... If \alpha\gamma is constant, as in equation 12, then \alpha varies with \gamma^{−1}." is difficult to follow. What is the intended conclusion here, and how does this impact the validity or interpretation of MR-GRIP?

10. While the authors explore several variations of their proposed framework, which is appreciated, many of these are introduced without sufficient background. It would benefit readers—especially those less familiar with these variations—if the authors provide more context or consider moving some of the more technical extensions to a well-structured Supplementary section.

[1] Lin, Zhaotong, Isaac Pan, and Wei Pan. "A practical problem with Egger regression in Mendelian randomization." PLoS genetics 18.5 (2022): e1010166.

Reviewer #3: The review is uploaded as an attachment

**Have all data underlying the figures and results presented in the manuscript been provided?**

Reviewer #1: Yes

Reviewer #2: None

Reviewer #3: Yes

PLOS authors have the option to publish the peer review history of their article (what does this mean? ). If published, this will include your full peer review and any attached files.

**Do you want your identity to be public for this peer review?** For information about this choice, including consent withdrawal, please see our Privacy Policy .

Reviewer #1: No

Reviewer #2: No

Reviewer #3: No

**Figure resubmission:**
---

## [Decision Letter · Decision Letter 1]

20 Oct 2025

PGENETICS-D-25-00735R1

Getting to GRIPS with MR-Egger: modelling directional pleiotropy independently of allele coding

PLOS Genetics

Dear Dr. Dudbridge,

Thank you for submitting your revised manuscript to PLOS Genetics. There are still some questions raised, especially some quite critical ones by Reviewer 1. Therefore, we invite you to submit a revised version of the manuscript that addresses the points raised during the review process.

Please submit your revised manuscript within 60 days . If you will need more time than this to complete your revisions, please reply to this message or contact the journal office at plosgenetics@plos.org. Please include the following items when submitting your revised manuscript:

We look forward to receiving your revised manuscript.

Kind regards,

Wei Pan

Guest Editor

PLOS Genetics

Xiaofeng Zhu

Section Editor

PLOS Genetics

Aimée Dudley

Editor-in-Chief

PLOS Genetics

Anne Goriely

Editor-in-Chief

PLOS Genetics

**Additional Editor Comments:**

I appreciate the authors' efforts in revising the paper according to the reviewers' comments. The revision was sent back to the original reviewers, two of whom are largely satisfied (with a few remaining and relatively minor questions/comments) while the other one seems to be more critical. I'd like to hear the authors responses, all or many of which perhaps can be incorporated in a revision (to help the reader with similar questions).

**Reviewers' comments:**

Reviewer's Responses to Questions

**Comments to the Authors:**

Reviewer #1: my review report is sent as an attachment.

Reviewer #2: The revised manuscript shows improvement compared to the previous version, but I still find certain key points lacking in clarity and logical justification. In particular, I would like to highlight the following issues that need further attention:

1. For point 2, I remain confused by this statement “the graphical interpretation of IVW … only make sense under all-positive coding”.

a. First, I suggest the authors clarify the motivation for introducing this discussion. Why is this statement conceptually or practically important to the argument being made?

b. Second, as far as I understand, the slope of each point remains the same regardless of the coding choice, and therefore the average slope should also remain invariant. If this is the case, why would the graphical interpretation only make sense under all-positive coding?

2. For point 9, my understanding is that this section aims to argue for the reasonableness of the VICE assumption. However, the explanation remains unclear and unconvincing.

a. The phrase “then under VICE we can only allow the following relationship” is vague and not sufficiently explained. It would help if the authors could explicitly define why it follows logically from the VICE assumption.

b. The explanations of inverse proportional to gamma, and of “stronger effect on exposure”, are difficult to follow and not strong enough. It would be great if the authors can provide clearer and stronger argument in this part, since I believe arguing the reasonableness of the VICE assumption is very important in this paper. Since the whole paper is built upon this assumption, the argumentation here needs to be much clearer and more rigorous.

Reviewer #3: Uploaded as an attachment

**Have all data underlying the figures and results presented in the manuscript been provided?**

Reviewer #1: Yes

Reviewer #2: None

Reviewer #3: Yes

PLOS authors have the option to publish the peer review history of their article (what does this mean? ). If published, this will include your full peer review and any attached files.

**Do you want your identity to be public for this peer review?** For information about this choice, including consent withdrawal, please see our Privacy Policy .

Reviewer #1: No

Reviewer #2: No

Reviewer #3: No

**Figure resubmission:**
---

## [Editor Report · Decision Letter 2]

21 Nov 2025

Dear Dr Dudbridge,

We are pleased to inform you that your manuscript entitled "Getting to GRIPS with MR-Egger: modelling directional pleiotropy independently of allele coding" has been editorially accepted for publication in PLOS Genetics. Congratulations!

Yours sincerely,

Wei Pan

Guest Editor

PLOS Genetics

Xiaofeng Zhu

Section Editor

PLOS Genetics

Aimée Dudley

Editor-in-Chief

PLOS Genetics

Anne Goriely

Editor-in-Chief

PLOS Genetics

BlueSky: @plos.bsky.social

Comments from the reviewers (if applicable):

I'd like to thank the authors' for their responses to the previous comments, and I am happy with the revision. Congratulations!

**Data Deposition**

http://datadryad.org/submit?journalID=pgenetics&manu=PGENETICS-D-25-00735R2

**Press Queries**

---

## [Editor Report · Acceptance letter]

PGENETICS-D-25-00735R2

Getting to GRIPS with MR-Egger: modelling directional pleiotropy independently of allele coding

Dear Dr Dudbridge,

We are pleased to inform you that your manuscript entitled "Getting to GRIPS with MR-Egger: modelling directional pleiotropy independently of allele coding" has been formally accepted for publication in PLOS Genetics! Your manuscript is now with our production department and you will be notified of the publication date in due course.

With kind regards,

Anita Estes

PLOS Genetics

On behalf of:
